# Arbuscular Mycorrhization Enhances Nitrogen, Phosphorus and Potassium Accumulation in *Vicia faba* by Modulating Soil Nutrient Balance under Elevated CO_2_

**DOI:** 10.3390/jof7050361

**Published:** 2021-05-05

**Authors:** Songmei Shi, Xie Luo, Xingshui Dong, Yuling Qiu, Chenyang Xu, Xinhua He

**Affiliations:** 1Centre of Excellence for Soil Biology, College of Resources and Environment, and Key Laboratory of Eco-Environments in Three Gorges Reservoir Region (Ministry of Education), School of Life Sciences, Southwest University, Chongqing 400716, China; shismei@email.swu.edu.cn (S.S.); luox123@email.swu.edu.cn (X.L.); xingshuid@outlook.com (X.D.); qyl19960101@email.swu.edu.cn (Y.Q.); xuchenyang6164@163.com (C.X.); 2National Base of International S&T Collaboration on Water Environmental Monitoring and Simulation in Three Gorges Reservoir Region, Chongqing 400716, China; 3School of Biological Sciences, University of Western Australia, Perth, WA 6009, Australia

**Keywords:** AMF, biomass production, carbon dioxide enrichment, nutrient absorption, soil nitrogen content, soil organic carbon

## Abstract

Effects of arbuscular mycorrhizal fungi (AMF), elevated carbon dioxide (eCO_2_), and their interaction on nutrient accumulation of leguminous plants and soil fertility is unknown. Plant growth, concentrations of tissue nitrogen (N), phosphorus (P), and potassium (K) in 12-week-old nodulated faba bean (*Vicia faba*, inoculated with *Rhizobium leguminosarum* bv. NM353), and nutrient use efficiency were thus assessed under ambient CO_2_ (410/460 ppm, daytime, 07:00 a.m.–19:00 p.m./nighttime, 19:00 p.m.–07:00 a.m.) and eCO_2_ (550/610 ppm) for 12 weeks with or without AM fungus of *Funneliformis mosseae* inoculation. eCO_2_ favored AMF root colonization and nodule biomass production. eCO_2_ significantly decreased shoot N, P and K concentrations, but generally increased tissue N, P and K accumulation and their use efficiency with an increased biomass production. Meanwhile, eCO_2_ enhanced C allocation into soil but showed no effects on soil available N, P, and K, while AM symbiosis increased accumulation of C, N, P, and K in both plant and soil though increased soil nutrient uptake under eCO_2_. Moreover, plant acquisition of soil NO_3_^−^–N and NH_4_^+^–N respond differently to AMF and eCO_2_ treatments. As a result, the interaction between AM symbiosis and eCO_2_ did improve plant C accumulation and soil N, P, and K uptake, and an alternative fertilization for legume plantation should be therefore taken under upcoming atmosphere CO_2_ rising. Future eCO_2_ studies should employ multiple AMF species, with other beneficial fungal or bacterial species, to test their interactive effects on plant performance and soil nutrient availability in the field, under other global change events including warming and drought.

## 1. Introduction

The atmospheric carbon dioxide (ACO_2_) is predicted to exceed 550 ppm by the end of this century from the 416 ppm in March 2021 (https://www.co2.earth, accessed on 1 May 2021). An elevated CO_2_ (eCO_2_) concentration has a direct effect on photosynthesis, and thus enhancements of carbon fixation and dry matter accumulation [1,2,3,4]. Shoot and root biomasses under eCO_2_ are increased by 20–30% across a wide range of crop species, such as wheat [5], rice [6], soybean [7], and tomato [8]. The beneficial effect of eCO_2_ on dry matter accumulation not only caused changes in C, nitrogen (N), phosphorous (P), and potassium (K) concentration, but also nutrient cycling from soil to plants [9]. When carbon fixation under eCO_2_ exceeds its ability to produce new sinks in plants, their photosynthetic rate decreases to balance source and sink capacity [2]. Moreover, increased biomass productivity of plants needs more supply of nutrients to match their increased carbon assimilation under eCO_2_ [9,10,11]. Soil nutrient availability might decrease over one-to-seven-year eCO_2_ exposure owing to an increased nutrient demand by eCO_2_-stimulated growth [12,13]. Decreases in soil nutrient supply modulate the magnitude of the eCO_2_ effect on plant biomass [14]. An ecosystem under eCO_2_ is not sustainable if no concomitant increases in N, P and K supply with the eCO_2_ stimulated growth can be maintained [15,16]. As a result, the availability of soil nutrients plays crucial roles in determining the response of plants to eCO_2_.

Arbuscular mycorrhizal fungi (AMF), the most widespread symbionts in nature, form obligate mutualistic associations with ~72% of terrestrial plant species [17]. Being dependent on host plant C, AMF received ~10–20% of recently photosynthetic C from host plants for their own growth. Meanwhile, AMF takes up and transfers significant amounts of N and P to the host plant, which improves plant growth, nutrient absorption, and water-use efficiency [18]. Numerous studies have shown that eCO_2_ exhibited a significant positive effect on C_3_ plants because of stimulated photosynthetic C assimilation, which increases the photosynthates transfer to roots of mycorrhizal plants, supporting symbiotic association [19,20,21]. In turn, the formation of AM generally increased leaf, stem, and root biomass production of the host plant under eCO_2_ [22,23,24].

Nutrient concentrations in C_3_ plant tissues generally decreased under eCO_2_ [15,25,26]. A meta-analysis of 7761 observations showed that average 689 ppm eCO_2_ decreased plant N, P, and K concentrations by 7%-15%, and N declined more than P and K [15,27]. However, a lower plant N, P, or K under eCO_2_ could be ameliorated by AM symbioses. For instance, compared to 50% decrease in non-AM plants, total plant P under 710 ppm eCO_2_ was only decreased by 22% in 56-d-old AM *Robinia pseudoacacia* [28]. Positive AMF effects on lettuce’s K concentration were greater under 700 ppm eCO_2_ than under ACO_2_ [29]. Chen et al. showed that^15^N and total N uptake by *Plantago lanceolata*, not by *Festuca arundinacea*, were enhanced by AMF inoculation under 730 ppm eCO_2_, suggesting that AMF effects on plant N uptake were species-specific under eCO_2_ [30]. Faba bean (*Vicia faba* L.), a leguminous food crop, associates with N_2_-fixing bacteria and alleviates N deficiency to some extent under eCO_2_ via symbiotic N_2_ fixation [2]. However, at present, the interactions effect of eCO_2_ and AMF on plant nutrition accumulation and its soil fertility in legumes remain unknown, which limits our understanding of the nutrient-limiting response of legume growth to eCO_2_. 

A constant daytime and nighttime ACO_2_ concentration or doubled-ACO_2_ concentrations have been applied for almost all studies involving the response of plants to future atmospheric CO_2_ concentrations. However, due to plant CO_2_ consumption and stronger wind turbulence, etc., at daytime, but higher soil and plant CO_2_ release, etc., at nighttime, the atmospheric CO_2_ concentrations at canopy height are usually higher at nighttime than at daytime in agricultural fields of Australia, Japan, and USA [31], and also in our onsite observation (2017–2019, 417 ± 16 ppm at daytime and 463 ± 27 ppm at nighttime) on the campus of Southwest University, Chongqing, China. As a consequence, plants should differentially respond to such contrasting daytime or nighttime atmosphere CO_2_ concentrations. Hence, supplying higher CO_2_ at nighttime than at daytime would presumably provide closer simulations of future atmospheric conditions. With comparisons to plants responding to different daytime and nighttime CO_2_ concentrations in environmentally controlled glass-made chambers, the objectives of the present study were to quantify (1) changes in C, N, P, and K accumulation in the plant–soil system of nodulated faba beans (inoculated with *Rhizobium leguminosarum* bv. NM353) under eCO_2_ and (2) capacity of AMF for enhancing growth and nutrient accumulation in fabas bean by facilitating nutrient uptake from soil under eCO_2_.

## 2. Materials and Methods

### 2.1. Experimental Setup 

The experiment was conducted in six temperature and humidity-controlled growth chambers (1.5 × 1 × 2.5 m) made by tempered glass (90% light transmission) at the National Monitoring Base for Purple Soil Fertility and Fertilizer Efficiency (29°48′ N, 106°24′ E, 266.3 m above the sea level) on the campus of Southwest University, Chongqing, China. The CO_2_ concentrations inside the growth chambers were also automatically maintained by a CO_2_ auto-controlling facility (DSS-QZD, Qingdao Shengsen Institute of CNC Technology, Shandong, China). The detailed information about the automatically controlled environment facility was described in our previous study [4]. The experimental pots (21 × 17 cm) were filled with 3.4 kg soil (Eutric Regosol, FAO Soil Classification System). The soil was air-dried, sieved by passing through a 2 mm mesh, and sterilized at 121 °C for 120 min. The texture of the soil was sandy loam with a pH 6.8, 10.56 g soil organic carbon kg^−1^, 0.66 g total N kg^−1^, 0.61 g total P kg^−1^, 97 mg available N kg^−1^, 17 mg available P kg^−1^, and 197 available K mg kg^−1^. 

### 2.2. Experimental Design and Treatments

The experiment was a split plot design with atmospheric CO_2_ concentrations as the main factor and mycorrhizal inoculation as the subfactor. The experiment involved two AMF inoculations (*Funneliformis mosseae*, AMF, and no *F**. mosseae* as control, non-AMF), and two CO_2_ treatments (current ACO_2_, 410/460 ppm, daytime, 07:00 a.m.–19:00 p.m./nighttime, 19:00 p.m.–07:00 a.m. and eCO_2_, 550/610 ppm as increased by ~33.33% of ACO_2_). Three growth chambers were used for ACO_2_ treatment, and the other three for eCO_2_ treatment. A total of six growth chambers were in a completed random arrangement.

The AMF inoculum (*Funneliformis mosseae*) was obtained from Bank of Glomales in China, located in the Beijing Academy of Agriculture and Forestry, Beijing. The inoculum had 50 spores per gram of dry soil in a mixture of soil, mycorrhizal mycelia, and root segments. Each AMF pot was supplied with 20 g *F. mosseae* inocula at 5 cm soil surface depth. Each non-AMF pot received an equal amount of autoclaved (121 °C, 0.1 Mpa, 120 min) inocula and 5 mL filtrate (0.45 µm syringe filter, Millipore Corporation, Billerica, MA, USA) from this AMF inoculum to minimize differences in microbial communities. Two AMF pots and two non-AMF pots were placed inside each growth chamber for a total of six replicated pots to each CO_2_ concentration treatment. Seeds of Faba bean (*Vicia faba* cv. 89–147) from Chongqing Academy of Agricultural Sciences were sterilized with 10% H_2_O_2_ for 20 min, rinsed with sterile water, and then pre-germinated on sterilized moist filter paper at 25/20 °C (day/night) for 60 h. The germinated seeds were soaked with liquid inoculate of *Rhizobium leguminosarum* bv. NM353 for 30 min and sown in each pot (four seeds per pot), and then 5 mL *R**. leguminosarum* bv. NM353 liquid inocula was also added to each pot. Except for the CO_2_ concentration, the chambers had similar growth conditions such as fertilization, light, air temperature, and humidity. The similar temperature and humidity between inside and outside the growth chambers were also automatically maintained by the above-mentioned CO_2_ auto-controlling facility [4]. The photosynthetic active radiation (PAR) was supplied by the natural light. To minimize differences in growth conditions, the position of growth pots in each chamber was rotated once a week and shifted to another replicate chamber once fortnightly. In addition, all the growth pots were watered once with Hoagland solution to a total of 100 mg N, 50 mg P, and 75 mg K per pot and were maintained at 70% water-holding capacity for the whole growth period. 

### 2.3. Harvest and Sampling 

Plant and soil samples were collected 12 weeks after sowing and were combined from two pots in each chamber as one composite sample. Thus, each treatment had three composite samples from six pots. Plant tissues were divided into shoots (leaves and stems) and roots. A portion of fresh roots was stored in 50% ethanol to determine root AM colonization. The remaining fresh roots, shoots, and nodules were dried at 105 °C for 30 min and then at 75 °C for > 48 h until at a consistent dry weight. Soil samples were air-dried for > 48 h for the determination of chemical properties. 

### 2.4. Mycorrhizal Colonization 

Root AMF colonization rate was determined after clearing roots in 10% (*w/v*) KOH and staining with 0.05% (*v/v*) trypan blue in lactic acid [32]. Briefly, 50 stained segments were randomly selected and mounted on microscope slides to assess mycorrhizal root colonization. The root segments were checked under a microscope at 200 magnification. Mycorrhizal colonization rate was calculated by the percentage of infected root segments numbers out of the total number of observed root segments. 

### 2.5. Determination of Carbon, Nitrogen, Phosphorus, and Potassium in Plants and Soils

The oven-dried shoot and root samples were ground into fine powder and then digested with 98% sulfuric acid and 30% hydrogen peroxide. Plant N concentrations were determined using the micro-Kjeldahl method, P concentrations using the vanadium molybdate yellow colorimetric method, and K concentrations by the flame photometry [33]. Carbon concentration was determined using the potassium dichromate-sulfuric acid oxidation method [33]. Soil available N was measured by the micro-diffusion technique after alkaline hydrolysis, and soil available P was extracted with 0.5 M NaHCO_3_ and measured by the Mo-Sb anti spectrophotometric method [33]. Soil available K was determined by the flame photometry after extracting with 1.0 M ammonium acetate [33]. Soil NO_3_^−^–N and NH_4_^+^–N concentrations were analyzed using a continuous flow analyzer (Seal Auto Analyzer III, Hamburg, Germany) according to the company’s manual after extracting with 1.0 M potassium chloride.

Plant tissue nutrient accumulation was calculated as tissue nutrient concentration multiplied with biomass production. The nutrient use efficiency (g) = (biomass production, g plant^−1^)^2^/(nutrient accumulation, mg plant^−1^) was expressed as biomass produced squared per unit of a certain nutrient accumulated [34]. 

### 2.6. Statistical Analysis

Data were statistically analyzed using the SPSS Statistics 19.0 (StatSoft Inc., Tulsa, USA). Results were presented as means ± SE (*n* = 3). Significant differences among treatments were compared by the Tukey’s Multiple Range Test at *P* < 0.05. Two-way analysis of variance (ANOVA) was conducted to determine the effects of CO_2_, AMF colonization, and their interactions. The Pearson’s correlation coefficients were calculated to assess the relationships between biomass productions and C, N, P, and K concentration in plant tissues and soils. 

## 3. Results

### 3.1. Mycorrhizal Colonization, Nodule Biomass, and Plant Growth Parameters 

Root colonization by *F. mosseae* was significantly increased by eCO_2_, while it was not detected in the plants with autoclaved *F. mosseae* inocula. Both AMF and eCO_2_ significantly increased nodule biomass, and the highest nodule biomass was observed in AMF plants grown under eCO_2_ (Table 1). 

eCO_2_ significantly increased shoot, root, and total plant biomass production and decreased root/shoot ratio, regardless of whether the faba beans were colonized by *F. mosseae* or not (*P* < 0.01, Table 1). Compared to the non-AMF plants, *F. mosseae* colonization increased shoot and total plant biomass production under both ACO_2_ and eCO_2_ (*P* < 0.01, Table 1), but not for root biomass production and root/shoot ratio (*P* > 0.05, Table 1). The interaction between CO_2_ and AMF was not significant, except for the nodule biomass production (*P* > 0.05, Table 1). 

### 3.2. Effects of AMF and CO_2_ on Plant Carbon, Nitrogen, Phosphorus and Potassium Concentrations

Shoot and root C concentrations were generally significantly higher under eCO_2_ than under ACO_2_ in both AMF and non-AMF plants (*P* < 0.01, Figure 1A,E), while they were unaffected by *F. mosseae* inoculation under the same CO_2_ (*P* = 0.74–0.76, Figure 1A,E). Except for shoot N concentrations under eCO_2_, shoot and root N concentrations were significantly greater in AMF than non-AMF plants under both ACO_2_ and eCO_2_ (*P* < 0.05–0.01, Figure 1B,F). Concentrations of shoot P or K in both non-AMF and AMF plants were significantly more decreased under eCO_2_ than under ACO_2_ (*P* < 0.0.05, Figure 1C,D), whereas root P or K concentrations were unaffected by either eCO_2_ or *F. mosseae* inoculation (Figure 1G,H). A significant CO_2_ × AMF interaction was observed in shoot C and N concentrations only (*P* < 0.05, Figure 1A,B), but not in shoot P and K concentrations or in root C, N, P, and K concentrations (Figure 1). 

### 3.3. Effects of AMF and CO_2_ on Plant Carbon, Nitrogen, Phosphorus, and Potassium Accumulations

Accumulations of C, N, P, and K in shoot, root, and total plant was increased more under eCO_2_ than under ACO_2_ in both AMF and non-AMF plants (*P* < 0.05, Figure 2). Compared with non-AMF plants, under eCO_2_, not under ACO_2_, AMF plants had greater shoot and total plant C, N, P, and K accumulations (*P* < 0.05, Figure 2A–D,I–L) and root N accumulations (*P* < 0.05, Figure 2F), but not root C, P, and K accumulations (*P* = 0.18–0.66, Figure 2E,G,H). In addition, no significant interactions of CO_2_ × AMF were observed, no matter whether plants were grown under different eCO_2_ or colonized by *F**. mosseae* (Figure 2). 

### 3.4. Effects of AMF and CO_2_ on Nutrient Use Efficiency

Compared to ACO_2_, nutrient use efficiency of N, P, and K was significantly increased under eCO_2_ in both non-AMF and AMF plants (Figure 3A–C). However, AMF colonization had no effect on N, P, and K use efficiency under the same CO_2_ concentration. When combined with eCO_2_ and AMF colonization, an increase in nutrient use efficiency of N, not P and K, was observed (*P* < 0.05, Figure 3A).

### 3.5. Effects of AMF and CO_2_ on Soil Nutrients 

eCO_2_ significantly increased soil organic carbon (*P* < 0.001, Figure 4A) and decreased soil NO_3_^−^–N (*P* < 0.01, Figure 4D), but had no effects on soil available N (*P* = 0.06, Figure 4B), available P (*P* = 0.06, Figure 4E), and available K (*P* = 0.37, Figure 4F) irrespective of non-AMF or AMF colonization. Soil NH_4_^+^–N was higher by 15.6% while lower by 19.7% under eCO_2_ than ACO_2_ in non-AMF and AMF soils, respectively (*P* < 0.05, Figure 4C). Meanwhile, AMF colonization significantly increased soil organic carbon, available N, NO_3_^−^–N, available P, and available K under both eCO_2_ and ACO_2_ (*P* < 0.01, Figure 4). In addition, significant interaction of CO_2_ × AMF was only in NH_4_^+^–N (*P* < 0.001, Figure 4C).

### 3.6. Correlations between Biomass Production and Plant Tissue or Soil Nitrogen, Phosphorus, and Potassiumconcentration 

Shoot biomass production was significantly positively correlated to shoot C concentration (R^2^ = 0.73, *P* < 0.05, Figure 5A), while negatively correlated to shoot N concentration in non-AMF plants (R^2^ = 0.76, *P* < 0.05, Figure 5D), but not in AMF plants (*P* > 0.05, Figure 5A,D). Shoot biomass production also significantly negatively correlated to shoot P or K concentration in both non-AMF and AMF plants (R^2^ = 0.55–0.85, *P* < 0.05, Figure 5G,J).

Root biomass production was significantly positively correlated to root C or N concentration in non-AMF plants (R^2^ = 0.64–0.78, *P* < 0.05, Figure 5B,E), but not in AMF plants (*P* = 0.27–0.51, Figure 5B,E), while significantly negatively correlated to root P concentration in AMF plants (R^2^ = 0.60, *P* < 0.05, Figure 5H), but not in non-AMF plants (*P* = 0.94, Figure 5H). No relationships between root biomass production and root K concentration were observed in both AMF and non-AMF plants (*P* = 0.41–0.45, Figure 5K). 

Total plant biomass production significantly positively correlated to soil organic carbon in both AMF (R^2^ = 0.60, *P* < 0.05) and non-AMF plants (R^2^ = 0.78, *P* < 0.05, Figure 5C), and also to soil available N, available P, or available K in AMF plants (R^2^ = 0.56–0.74, *P* < 0.05), but not in non-AMF plants (*P* = 0.22–0.54, Figure 5F,I,L).

## 4. Discussion

### 4.1. Mycorrhizal Colonization and Nodule Biomass Increased under eCO_2_

The percentage of root colonization of faba beans by *F. mosseae* was significantly increased under eCO_2_ (Table 1), in agreement with previous studies [20,24,35]. Compared to 336–400 ppm ACO_2_, a meta-analysis from 434 observations demonstrated that 23% of extraradical hyphal length and 22% of mycorrhizal fungal biomass were increased under 550–1000 ppm eCO_2_ [36]. The increased AMF colonization might be closely linked to an increased C allocation to their external hyphae and an enlarged root biomass production when plants were grown under eCO_2_ [37,38,39]. Moreover, the nodule biomass under eCO_2_ was significantly increased in both non-AMF and AMF plants (Table 1), and higher in AMF than in non-AMF plants (Table 1), suggesting that AMF did promote nodulation or N_2_ fixation [40]. An improved photosynthesis under eCO_2_ resulted in an increased translocation of photosynthates from shoots to roots [23,41,42], thus also favoring nodule development (Table 1). In turn, AMF and N_2_-fixing bacteria received a certain amount of photosynthetic C from host plants to maintain their symbiotic interactions [2,18]. A similar nodule biomass in non-AMF and AMF plants under ACO_2_ (Table 1) might be due to competitions for plant carbohydrate between AMF and N_2_-fixing bacteria [40,41]. AMF was the dominant symbiont for photosynthetic C sink in the dual AMF/N2-fixing bacterial symbiosis [40]. Once AMF colonization reached the plateau phase, more C was available for nodule growth [43] and consequently higher nodule biomass in AMF than in non-AMF plants under eCO_2_. 

### 4.2. AMF and eCO_2_ Synergistically Improve Carbon Accumulation and Biomass Production

Studies have reported that AMF can promote plant growth under eCO_2_ due to enhanced nutrient uptake and photosynthetic rate of the host plant [20,21,23]. Baslam et al. [42] found that both 700 ppm eCO_2_ and AMF symbiosis increased shoot and root biomass production in vegetative nodulated alfalfa. Our results also showed that 550/610 ppm eCO_2_ significantly improved shoot, root, and total plant biomass production (Table 1). Greater root biomass could be essential to sustain increased shoot biomass, since roots might be less efficient in transferring nutrients due to reduced transpiration-driven mass flow under eCO_2_ [44]. For example, a greater effect of 700 ppm eCO_2_ on faba bean roots at the stem elongation stage was observed in well-watered than in drought conditions [2]. However, the present results showed that eCO_2_ generally stimulated more shoot biomass than root biomass production, leading to a significant decrease of root/shoot ratios in both AMF and non-AMF plants (Table 1). This finding agreed with results of Baslam et al. [21] in non-AMF alfalfa that 700 ppm eCO_2_ increased the shoot to root ratios in both seven-week-old and nine-week-old plants, while all plants gradually increased their biomass partitioning to roots over the vegetative growth. Butterly et al. [26] showed 550 ppm eCO_2_ significantly decreased the root–shoot ratio of field pea and wheat at grain filling but not at maturity [26]. These results suggested changes in biomass partitioning into roots under eCO_2_ varied with plant growth stages. These alterations might be caused by the internal balance between labile N and C in the shoot and root [45]. A higher investment of photosynthates to roots could maintain better supply of nutrients through a well-developed root system for eCO_2_ induced photosynthesis [46].

Shoot and root biomass production were significantly positively correlated to C concentrations in non-AMF plants (R^2^ = 0.64–0.73, *P* < 0.05, Figure 5A,B), indicating that an enhanced growth under eCO_2_ was most likely due to an improved net photosynthetic rate [2]. AMF are known to receive photosynthetic C from host plants, and more C was available to roots under eCO_2_ [24], leading to the enhanced photosynthesis rate as a consequence of the increased C sink strength in mycorrhizal plants [40]. Such effects of mycorrhizal symbioses should reduce plant photosynthesis acclimation caused by carbohydrate accumulation when they were grown under long-term exposure to eCO_2_ [47]. However, Gavito et al. [48] found no evidence of photosynthetic acclimation in *F. caledonium* pea grown under 700 ppm eCO_2_ for nine weeks. In a greenhouse study, Goicoechea et al. [23] observed that *R. intraradices* association accelerated photosynthetic acclimation of alfalfa at the end of the vegetative stage under 700 ppm eCO_2_. The C flow from host to AMF in roots might depend on the developmental stage of the different sinks within plants [49] or differences among AMF taxa in their exchange of carbon and nutrients [50]. Therefore, the effects of interactions between AMF and eCO_2_ on photosynthetic acclimation, C accumulation, and dynamics merit further investigations.

### 4.3. AMF and eCO_2_ Synergistically Improve Nitrogen Accumulation in Plant

It has been reported frequently that eCO_2_ affects nutrient concentrations due to photosynthesis improvement, which was associated with the enhancement of the carboxylation reaction of RuBisco [51]. Supporting such a concept, our results showed 11.8% of shoot N concentrations were decreased by eCO_2_ in AMF plants, but not in non-AMF plants (Figure 1B). Bloom et al. [52] suggested that eCO_2_ inhibited N assimilation and decreased N concentration in wheat and *Arabidopsis* plant. However this could not explain the reduction of shoot N concentration in AMF plants only (Figure 1B). The negative relationship between biomass production and shoot N concentration in AMF plants (Figure 5D) indicated that the reduced N might be ascribed to the increase in C assimilation and growth of AMF plants corresponding to plant N uptake, thereby reducing N concentration due to N dilution [24]. Moreover, we speculated that the mismatch between N demand and N assimilation by faba beans was overcome by a higher N uptake in a more developed root system under eCO_2_. This was evidenced by a respective 12% and 16% increase of root N concentrations in non-AMF and AMF plants under eCO_2_ (Figure 1F) and a significant relationship between root biomass production and root N concentrations (R^2^ = 0.78, *P* < 0.05, Figure 5E). The increased root biomass and root N concentration under eCO_2_ might lead to a higher proportion in root and a lower proportion of plant N in shoot. Drake et al. [53] suggested that the redistribution of N due to RuBisco acclimation under eCO_2_ could greatly increase N use efficiency. As the C assimilation was greater under eCO_2_ than under ACO_2_ (Figure 2A,E,I), the N, P, and K use efficiency was markedly increased in response to eCO_2_ (Figure 3), while the increase in nutrient uptake rate might not keep up with biomass increase [34,54]. Nevertheless, nutrient use efficiency could be enhanced by each or all of the following three pathways: (i) biomass increase and constant nutrient absorption, (ii) constant biomass and lower absorption of nutrients, and (iii) increase in nutrient absorption rate being lower than that in dry mass increase rate [54]. 

Additionally, enhanced N accumulations in shoot, root, and total plant were observed under eCO_2_ due to an increased biomass production (Figure 2B,F,J). Thus, plant N demand for C assimilation under eCO_2_ was not reduced in the present study. AMF also significantly improved plant N accumulation regardless CO_2_ level (Figure 2B,F,J). Similarly, a range of 30% to 41% increased N accumulation under 700–1000 ppm eCO_2_ has been reported in AMF colonized alfalfa [23], *Plantago lanceolate* [30], *Taraxacum officinale* [20], and wheat [24]. Fellbaum et al. [55] reported that an increase of C supply to the host plant could stimulate N transport in AM symbiosis. In turn, AMF affected C translocation and dynamics in plant–soil systems [56]. Thus, AMF may regulate the demand for C and the supply for N to the host plant under eCO_2_, and thus alter the balance of C and N. 

### 4.4. AMF and eCO_2_ Synergistically Improve Phosphorus and Potassium Accumulation in Plants

Concentrations of P and K under 550–750 ppm eCO_2_ were increased in lettuce [29], *Oryza sativa*, and *Echinochloa crusgalli* [57], while they were reduced in *Artemisia annua* [58], tomatoes [8], and durum wheat [59], but not significantly changed in oregano [22] and mung bean [60]. Our results showed that shoot P and K concentrations were significantly decreased under 550/610 ppm eCO_2_ for both non-AMF and AMF faba beans (Figure 1C,D). Moreover, a significantly negative linear correlation between shoot dry biomass and P or K concentrations (R^2^ = 0.55–0.85, *P* < 0.05, Figure 5G,J) was observed, suggesting that P or K concentrations were diluted by the increased shoot biomass production and such dilution effect also occurred in various C_3_ plants under varied eCO_2_ [15,19]. In addition, lower stomatal conductance and transpiration rates under eCO_2_ might limit the transpiration-driven mass flow of nutrients through soil to root, thereby lowering the absorption of N, P, K, calcium, magnesium, and sulphur [45,61,62]. 

It is well known that AMF play important roles in the acquisition of essential nutrients including N, P, K, and other elements by through hyphal translocation [18]. Lower concentrations of N, P, magnesium, sulphur, zinc, and manganese in lettuce [29], tomato [8], and wheat [59] under 700–970 ppm eCO_2_ have been shown to be mitigated by AMF inoculation. Compared to the non-mycorrhizal controls, significantly higher P and K accumulation in shoot and total plant displayed in the *F. mosseae* colonized faba bean regardless the CO_2_ concentrations applied (Figure 2C,D,K,L), suggesting that AMF enhanced plant uptake of P and K. The higher biomass production and consequently greater nutrients demand of AMF plants under CO_2_ might partly explain their increased nutrient accumulations. Similarly, Olesniewicz and Thomas [28] observed a significant increase of plant N and P in mycorrhizal *Robinia pseudoacacia* under 710 ppm eCO_2_. In a study with two citrus spp., *Rhizophagus intraradices* colonization under 700 ppm eCO_2_ stimulated plant growth and P acquisition of sour orange (*Citrus aurantium*), but not of sweet orange (*C. sinensis*) [63]. For *Pisum sativum*, however, the positive impact of AMF on shoot and total P disappeared under 700 ppm eCO_2_ [48]. Therefore, effects of AMF inoculation on plant nutrient uptake could be species-specific at high CO_2_ concentration. 

4.5. eCO_2_ has no Effect on Soil Nitrogen, Phosphorus, and Potassium Availability, but AMF Improves Soil Properties

Almost all eCO_2_ studies have focused on plant characteristics and less on the feedback on the plant nutrient uptake and soil nutrients supply [12,13,56]. It is well-known that plant growth depends mainly on the nutrient reserves in soil [64], and eCO_2_ might cause unpredictable repercussions on nutrient dynamics between soil nutrient pool and plant requirement [57]. When plants are grown under eCO_2_, soil nutrient availability could decrease over longer time periods because of greater dry weights and more nutrient demands, resulting in a lower eCO_2_ stimulation effect [65]. Our results showed that soil available N, available P, and available K concentrations were not significantly changed under eCO_2_ (Figure 4B,E,F), suggesting that soil nutrients were not a limitation. The lack of response of soil nitrogen and phosphorus to eCO_2_ (Figure 4B,E) and significantly increased nodule biomass under eCO_2_ (Table 1) implied that soil P availability did not limit N fixation in the present study and the increased N_2_ fixation could meet the part of plant N demand. However, it is still unknown that how much N could be fixed by N_2_-fixing bacteria under eCO_2_ and/or AMF. The determination of N_2_-fixing could thus provide the understanding of the dynamics behind the positive impacts of AMF and eCO_2_ on leguminous plants. 

Furthermore, the present results showed that significantly higher soil organic carbon, available N, available P, and available K (Figure 4) were in AMF than in non-AMF soils, especially under eCO_2_. This higher increase in soil organic carbon might be attributed to higher biomass production under eCO_2_ [59]. eCO_2_ allowed greater C investment to AMF and stimulated their biomass and activity [36,56]. In contrast, AMF increased the CO_2_ fixation and induced the transport of carbon from plant to soil, thereby increasing soil organic carbon content [66]. A generally increase trend in soil organic carbon with AMF colonization was also synchronized in rice grown under 550 and 700 ppm eCO_2_ [57]. Moreover, an increased AMF response to eCO_2_ could result in increasing decomposition of complex organic material [66], and the increasing organic matter decomposition might hence accelerate soil nutrient mineralization rate, resulting in a better N and P availability to plants [67,68,69]. Therefore, it may not be necessary to apply more N, P, and K fertilizers to soil for faba beans’ growth at their vegetative growth stage under eCO_2_.

Although soil available N concentration was unresponsive to eCO_2_, soil NO_3_^−^–N and NH_4_^+^–N were affected by either eCO_2_ or AMF (Figure 4C,D). AMF can obtain both NH_4_^+^–N and NO_3_^−^–N and then transfer them to host plants [70]. Uptake of N in the form of NO_3_^−^–N by AMF may play a major role in agricultural soils, where NO_3_^−^–N is often the dominant N form [52]. In line with this, soil NO_3_^−^–N was significantly higher in AMF than in non-AMF soils under both ACO_2_ and eCO_2_ (Figure 4D). In contrast, eCO_2_ decreased soil NO_3_^−^–N in both AMF and non-AMF soils (Figure 4D), but increased NH4^+^–N in non-AMF soil only (Figure 4C). The decrease in soil NO_3_^−^–N under eCO_2_ was consistent with the result in poplar, in which 550 ppm eCO_2_ negatively influenced nitrification by 44% under unfertilized treatment and consequently decreased NO_3_^−^–N availability by 30% on average [71]. Meanwhile, ~560 ppm eCO_2_ reduced the population of ammonia-oxidizing bacteria in soil and nitrification activities, leading to an increase of soil NH_4_^+^–N and a decrease of the NO_3_^–^–N [72]. Therefore, the CO_2_-stimulated increase on biomass may be attributed to an increased uptake of NO_3_^−^–N by the faba bean, thus depleting soil NO_3_^−^–N pool, whereas the lower NH_4_^+^–N observed under eCO_2_ in AMF soils (Figure 4C) showed that a high plant demand for NH_4_^+^ might be the primary driver for CO_2_ enhancement of AMF-mediated organic matter decomposition [66].

## 5. Conclusions

Although eCO_2_ generally resulted in reduction of shoot N, P, and K concentration, the nutrient use efficiency and accumulation of N, P, and K in shoot, root, and total plant were generally significantly increased. Meanwhile, eCO_2_ enhanced C allocation into soil but did not reduce soil available N, P, and K. Compared to non-AMF treatment, AM symbiosis increased their accumulation of C, N, P, and K in both plant and soil though increasing organic matter decomposition and soil nutrient uptake under eCO_2_. Moreover, soil NO_3_^–^–N under eCO_2_ was significantly higher, but soil NH_4_^+^–N was lower in AMF than in non-AMF soils, suggesting that plant acquisition of soil NO_3_^−^–N and NH_4_^+^–N responded differently to AMF and eCO_2_ treatments. As a result, the interaction between AM symbiosis and eCO_2_ did improve plant C accumulation and soil N, P, and K uptake in faba bean, and an alternative fertilization programs for legume plantation should be hence taken under upcoming atmosphere CO_2_ rising scenarios. Further eCO_2_ studies should employ two or more mycorrhizal fungal species, either individual or multi-species combined, also with other fungal or bacterial species, to test their interactive effects on both plant growth and yield production, and soil nutrient availability in the field, under other global environment change scenarios, such as warming or temperature rising, drought, and so on.

## Figures and Tables

**Figure 1 jof-07-00361-f001:**
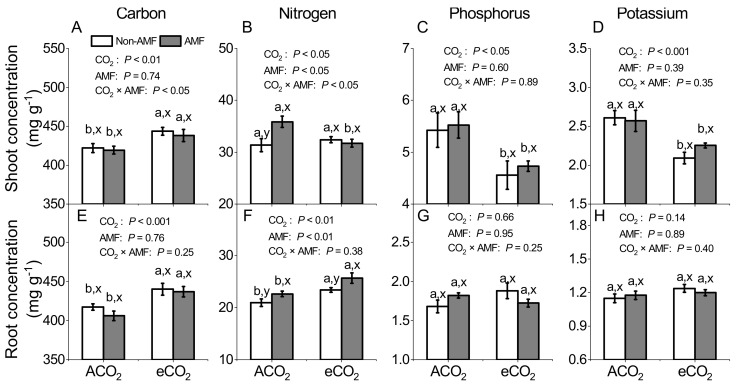
Concentrations of carbon (**A**,**E**), nitrogen (**B**,**F**), phosphorus (**C**,**G**), and potassium (**D**,**H**) in shoots and roots of *Funneliformis mosseae* inoculated faba beans grown for 12 weeks at ambient CO_2_ (ACO_2_) and elevated CO_2_ (eCO_2_). Data (means ± SE, *n* = 3) followed by different letters indicate significant differences between CO_2_ treatments for the same AMF inoculation (a, b) and between AMF inoculations for the same CO_2_ treatment (x, y) at *P* < 0.05 as revealed by Tukey’s test. Statistical comparisons (two-way ANOVA) between AMF or CO_2_ treatments as well as their interaction (AMF × CO_2_) are presented for each variable.

**Figure 2 jof-07-00361-f002:**
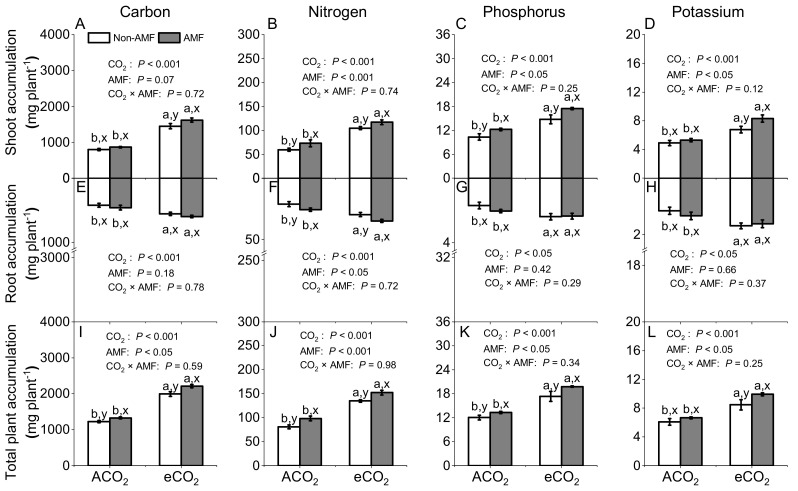
Accumulations of carbon (**A**,**E**,**I**), nitrogen (**B**,**F**,**J**), phosphorus (**C**,**G**,**K**), and potassium (**D**,**H**,**L**) in shoots, roots, and total plants of *Funneliformis mosseae* inoculated faba beans grown for 12 weeks at ambient CO_2_ (ACO_2_) and elevated CO_2_ (eCO_2_). Data (means ± SE, *n* = 3) followed by different letters indicate significant differences between CO_2_ treatments for the same AMF inoculation (a, b) and between AMF inoculations for the same CO_2_ treatment (x, y) at *P* < 0.05 as revealed by Tukey’s test. Statistical comparisons (two-way ANOVA) between AMF or CO_2_ treatments as well as their interaction (AMF × CO_2_) are presented for each variable.

**Figure 3 jof-07-00361-f003:**
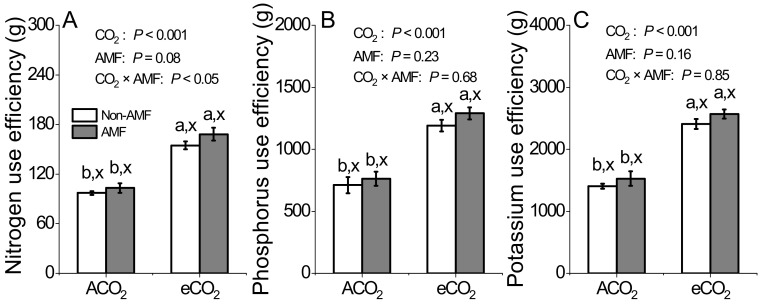
Nirogen (**A**), phosphorus (**B**), and potassium (**C**) use efficiency in the total plants of *Funneliformis mosseae* inoculated faba beans grown for 12 weeks at ambient CO_2_ (ACO_2_) and elevated CO_2_ (eCO_2_). Data (means ± SE, *n* = 3) followed by different letters indicate significant differences between CO_2_ treatments for the same AMF inoculation (a,b) and between AMF inoculations for the same CO_2_ treatment (x, y) at *P* < 0.05 as revealed by Tukey’s test. Statistical comparisons (two-way ANOVA) between AMF or CO_2_ treatments as well as their interaction (AMF CO_2_) are presented for each variable.

**Figure 4 jof-07-00361-f004:**
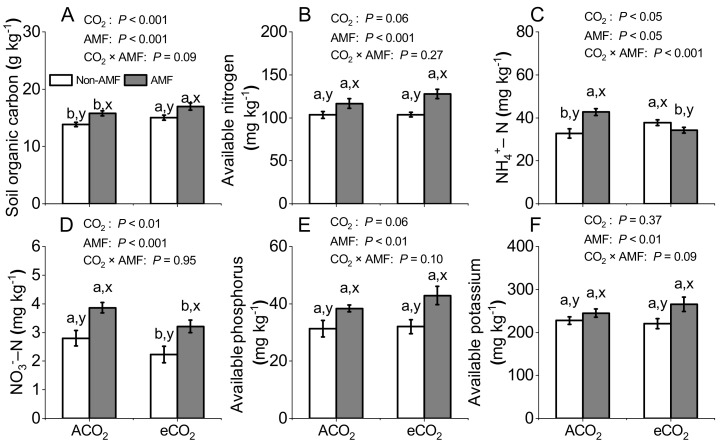
Soil organic carbon (**A**), soil available N (**B**), soil NH_4_^+^–N (**C**), soil NO_3_^−^ –N (**D**), available phosphorus (**E**), and soil available potassium (**F**) in soils of *Funneliformis mosseae* inoculated faba beans grown for 12 weeks at ambient CO_2_ (ACO_2_) and elevated CO_2_ (eCO_2_). Data (means ± SE, *n* = 3) followed by different letters indicate significant differences between CO_2_ treatments for the same AMF inoculation (a, b) and between AMF inoculations for the same CO_2_ treatment (x, y) at *P* < 0.05, as revealed by Tukey’s test. Statistical comparisons (two-way ANOVA) between AMF or CO_2_ treatments as well as their interaction (AMF × CO_2_) are presented for each variable.

**Figure 5 jof-07-00361-f005:**
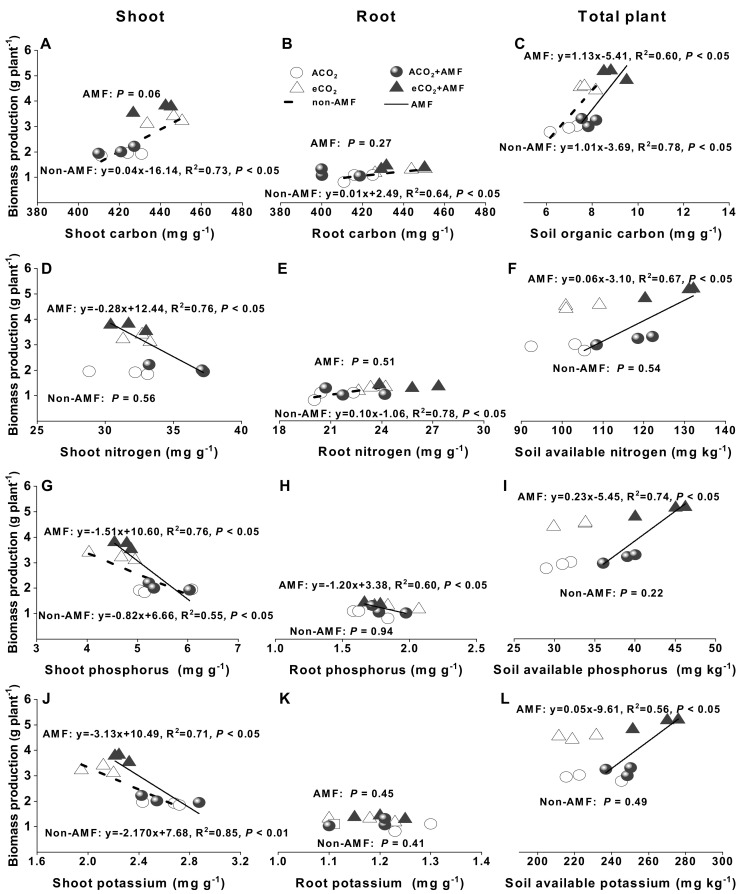
Relationships between carbon (**A**–**C**), nitrogen (**D**–**F**), phosphorus (**G**–**I**), and potassium (**J**–**L**) concentration in shoots, roots, and soils and plant biomass production of *Funneliformis mosseae* inoculated faba beans grown for 12 weeks at ambient CO_2_ (ACO_2_) and elevated CO_2_ (eCO_2_). Open circles and triangles represent data under ACO_2_ and eCO_2_ in non-AMF plants, and closed circles and triangles represent data under ACO_2_ and eCO_2_ in AMF plants, respectively. Regressions are shown for non-AMF (dotted lines) and for AMF (solid lines) treatments, *n* = 6.

**Table 1 jof-07-00361-t001:** Percentages of arbuscular mycorrhizal (AM) colonization, nodule biomass production, shoot biomass production, root biomass production, root/shoot ratio, and total biomass production in *Funneliformis mosseae* inoculated faba beans grown for 12 weeks at ambient CO_2_ (ACO_2_) and elevated CO_2_ (eCO_2_).

Treatment		AM (%)	Nodule Biomass (g plant^−1^)	Shoot Biomass (g plant^−1^)	Root Biomass (g plant^−1^)	Root/Shoot Ratio	Total Biomass (g plant^−1^)
ACO_2_	M^−^	0	0.04 ± 0.01b,x	1.91 ± 0.04b,y	1.01 ± 0.09b,x	0.53 ± 0.06a,x	2.91 ± 0.07b,y
	M^+^	43.01 ± 2.49b	0.05 ± 0.01b,x	2.15 ± 0.08b,x	1.13 ± 0.08b,x	0.57 ± 0.05a,x	3.18 ± 0.09b,x

eCO_2_	M^−^	0	0.07 ± 0.00a,y	3.24 ± 0.09a,y	1.26 ± 0.05a,x	0.39 ± 0.01b,x	4.50 ± 0.08a,y
	M^+^	59.64 ± 3.04a	0.13 ± 0.02a,x	3.69 ± 0.09a,x	1.39 ± 0.14a,x	0.37 ± 0.01b,x	5.05 ± 0.12a,x
ANOVA							
CO_2_		**	***	***	**	**	***
AMF			*	**	ns	ns	**
CO_2_ × AMF			**	ns	ns	ns	ns

Data (means ± SE, *n* = 3) followed by different letters indicate significant differences between CO_2_ treatments for the same AMF inoculation (a, b) and between AMF inoculations for the same CO_2_ treatment (x, y) at *P* < 0.05 as revealed by Tukey’s test. ANOVA: ns not significant; *, **, and *** significant at *P* ≤ 0.05, *P* ≤ 0.01, and *P* ≤ 0.001, respectively. M^+^ or M^–^, with or without *F. mosseae* inoculation.

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
