# Peer review of "Arbuscular Mycorrhization Enhances Nitrogen, Phosphorus and Potassium Accumulation in Vicia faba by Modulating Soil Nutrient Balance under Elevated CO2"

_jof, 2021, doi:10.3390/jof7050361_

Round 1
Reviewer 1 Report
In this paper, the authors investigate the relationship between elevated atmospheric CO2 concentrations and mycorrhizal symbiosis and nutrition of Vicia faba. As the authors point out, an increase in atmospheric CO2 concentration may increase plant photosynthesis and soil organic matter deposition, but it may also change the amount of soil nutrients and their absorption by plants. The relationship between plant physiology and global environmental change will be an important area of research in the future, and the accumulation of knowledge will be significant for efficient crop production and its sustainability. There are no major problems with the design of the experiment, the interpretation of the results, or the discussion of them. The data is discussed objectively and the paper is well written. However, several studies of this kind have been done in the past, including on other plant species, and this study does not examine a completely new hypothesis. In this study, only one AMF was used. It is often said that we need to adapt the nutritional and physiological characteristics of plants in response to climate change, but how the modification of soil microbes affects plant growth and the perspective of breeding soil microbes is also an interesting topic.Of course, it is difficult to breed microorganisms, but if plant physiology changes significantly in response to climate change, then the physiological and genetic characteristics of microorganisms that also change in response to the change should also be understood. Since this study only discusses interactions with specific mycorrhizal fungi, I think the limitations of this and the future perspective of the study should also be mentioned in the discussion.
Author Response
In this paper, the authors investigate the relationship between elevated atmospheric CO2 concentrations and mycorrhizal symbiosis and nutrition of Vicia faba. As the authors point out, an increase in atmospheric CO2 concentration may increase plant photosynthesis and soil organic matter deposition, but it may also change the amount of soil nutrients and their absorption by plants. The relationship between plant physiology and global environmental change will be an important area of research in the future, and the accumulation of knowledge will be significant for efficient crop production and its sustainability. There are no major problems with the design of the experiment, the interpretation of the results, or the discussion of them. The data is discussed objectively and the paper is well written.
Response: Thanks for your comment.
However, several studies of this kind have been done in the past, including on other plant species, and this study does not examine a completely new hypothesis. In this study, only one AMF was used. It is often said that we need to adapt the nutritional and physiological characteristics of plants in response to climate change, but how the modification of soil microbes affects plant growth and the perspective of breeding soil microbes is also an interesting topic. Of course, it is difficult to breed microorganisms, but if plant physiology changes significantly in response to climate change, then the physiological and genetic characteristics of microorganisms that also change in response to the change should also be understood. Since this study only discusses interactions with specific mycorrhizal fungi, I think the limitations of this and the future perspective of the study should also be mentioned in the discussion.
Response: Thanks for your comment. Yes, the future studies should employ two or more mycorrhizal fungal species, either individual or multi-species combined, and also with other fungal or bacterial species, to test their interactive effects on both plant growth and yield production, and changes in soil nutrient availability in thefield, under global environment change scenarios, such as warming or temperature rising, drought and other extreme events.
Reviewer 2 Report
Dear authors,
I appreciate the opportunity to review the article entitled: "Arbuscular mycorrhizal colonization facilitate carbon, nitrogen, phosphorus and potassium accumulation in Vicia faba by modulating soil nutrient balance under elevated atmospheric CO2".
Overall is a very well-written article, but some information can be clarified and further discussed. Please see the attached PDF for details.

Author Response
- The title is too long. Consider reviewing it.
Response: Thanks for your comment. The previous title of “Arbuscular mycorrhizal colonization facilitated carbon, nitrogen, phosphorus and potassium accumulation in Vicia faba by modulating soil nutrient balance under elevated atmospheric CO2” has been change to “Arbuscular mycorrhization enhances nitrogen, phosphorus and potassium accumulation in Vicia faba by modulating soil nutrient balance under elevated CO2”
- Line 42-43 Rewrite or clarify this sentence.
Response: Thanks for your comment. This sentence “On the one hand, the C fixation in plants under eCO2 exceeds the capacity to produce new sinks, the plants decrease their photosynthetic rate to balance source activity and sink capacity” has been changed to “When carbon fixation under eCO2 exceeds their ability to produce new sinks in plants, their photosynthetic rate would decrease to balance source and sink capacity [2].”
- Line 44-45 Why? Clarify.
Response: Thanks for your comment. Because increased biomass productivity of plants needs more supply of nutrients to match their increased carbon assimilation under eCO2 [9-11]. Soil nutrient availability might decrease over 1-7 year eCO2 exposure owing to an increased nutrient demand by eCO2-stimulated growth [12-13]. Decreases in soil nutrient supply modulate the magnitude of the eCO2 effect on plant biomass [14].
- Line 51 Update the information and the reference.
Response: Thanks for your comment. The information has been change to “Arbuscular mycorrhizal fungi (AMF), the most widespread symbionts in nature, form obligate mutualistic associations with ~72 % of terrestrial plant species [17] (see below)”.
Brundrett, M. C., Tedersoo, L., Evolutionary history of mycorrhizal symbioses and global host plant diversity. New Phytol. 2018, 220(4), 1108–1115.
- Line 57 Add citation.
Response: Thanks for your comment. The citations “Reference 19, 20 and 21” have been added in the revised reversion.
- Line 63-65These two phrases do not make sense.
Response: Thanks for your comment. The previous sentences of “However, a lower concentration of N, P or K in plant tissues under eCO2 could be ameliorated by the AM symbioses. For instance, eCO2 at 710 ppm resulted in a 50 % or 22 % decrease in total plant P in 56-d-old non-mycorrhizal or mycorrhizal Robinia pseudoacacia [20]” has been changed to “However, a lower plant N, P or K under eCO2 could be ameliorated by AM symbioses. For instance, compared to 50 % decrease in non-AM plants, total plant P under 710 ppm eCO2 was only decreased by 22 % in 56-d-old AM Robinia pseudoacacia [28]”.
- Line71 Add citation.
Response: Thanks for your comment. The citation “Reference 2” has been added in the revised reversion.
- Line 112 Two AMF?
Response: Thanks for your concern, which is also commented by the reviewer 3. The previous sentence of “The experiment involved two AMF (with Funneliformis mosseae and without as control)” has been changed to “The experiment involved two AMF inoculations (Funneliformis mosseae, AMF and no F. mosseae as control, non-AMF)”.
- Line 122 Clarify this information.
- Line 123-125 Clarify this information.
Response: Thanks for your comment. The sentences of Line 122-125 “Non-autoclaved or autoclaved (0.1 Mpa, 121 ℃, 120 min) F. mosseae inocula (20 g) were supplied at 5 cm soil surface depth inside each AMF or non-AMF pot. 5 mL filtrate (0.45 μm syringe filter, Millipore Corporation, Billerica, MA, USA) from the AMF inoculum was added to each non-AMF pot to minimize differences in microbial communities” have been change to “Each AMF pot were supplied with 20 g F. mosseae inocula at 5 cm soil surface depth. Each non-AMF pot received an equal amount of autoclaved (121 ℃, 0.1 Mpa, 120 min) inocula and 5 mL filtrate (0.45 µm syringe filter, Millipore Corporation, Billerica, MA, USA) from this AMF inoculum to minimize differences in microbial communities.”
- Line128 Where the cultivar is from?
Response: Thanks for your comment. The sentence of “Faba bean (Vicia faba cv. 89-147) seeds were sterilized with 10 % H2O2 for 20 min” has been change to “Seeds of Faba bean (Vicia faba cv. 89-14), from Chongqing Academy of Agricultural Sciences were sterilized with 10 % H2O2 for 20 min”.
- Line 130 I believe that the information on inoculation with rhizobium should be clarified in the abstract and introduction sections.
- Line132 liquid inoculum of AMF?
Response: Thanks for your comment. The liquid inoculum was Rhizobium leguminosarum bv. NM353 please see Lines 19 of the abstract and Lines 95 of introduction sections in the revised version.
- Line 256 Explain abbreviation of SOC
Response: Full term of “soil organic carbon” has been now used, thanks..
- Line 307-310 Elaborate this further and add other citations.
Response: Thanks for your comment. The writings of “In turn, the presence of N2-fixing bacteria could also boost the photosynthesis rate as a consequence of more N supply through N2 fixation and the increased C sink strength due to C costs of N2 fixation. A similar nodule biomass in non-AMF and AMF plants under ACO2 (Table 1) might be due to competitions for plant carbohydrate between AMF and N2-fixing bacteria [36]” have been changed to “ In turn, AMF and N2-fixing bacteria received a certain amount of photosynthetic C from host plants to maintain their symbiotic interactions [2, 18]. A similar nodule biomass in non-AMF and AMF plants under ACO2 (Table 1) might be due to competitions for plant carbohydrate between AM fungi and N2-fixing bacteria [41-42]. AMF was the dominant symbiont for photosynthetic C sink in the dual AMF/N2-fixing bacterial symbiosis [41] Once AMF colonization reached the plateau phase, more C was available for nodule growth [44], consequently higher nodule biomass in AMF than in non-AMF plants under eCO2.”
- Line314 studies?
Response: Thanks for your comment. The more citations “Reference 20, 21 and 23” have been added in the revised reversion.
- Line 323-325 Discuss further the implications for the plant development.
Response: Thanks for your comment. The writings of “However, the present results showed that eCO2 generally stimulated shoot biomass more than root biomass, leading to a significant decrease of root/shoot ratios in both AMF and non-AMFplants (Table 1).Similar to our results, Butterly et al. (2015) showed 550 ppm eCO2 sig nificantly decreased root-shoot ratio of field pea at grain filling but not at maturity under 40 or 100 mg N kg-1 [18]. The partitioning of biomass into shoots and roots was determined by the internal balance between labile N and C in the shoot and root systems under different growth stages [39].” have been changed to “However, the present results showed that eCO2 generally stimulated more shoot biomass than root biomass production, leading to a significant decrease of root/shoot ratios in both AMF and non-AMF plants (Table 1). This finding agreed with results of Baslam et al. [21] in non-AMF alfalfa that 700 ppm eCO2 increased the shoot to root ratios in both 7-week-old and 9-week-old plants, while all plants gradually increased their biomass partitioning to roots over the vegetative growth. Butterly et al. [26] showed 550 ppm eCO2 significantly decreased root-shoot ratio of field pea and wheat at grain filling but not at maturity [26]. These results suggested changes in biomass partitioning into root under eCO2 varied with plant growth stages. These alterations might be caused by the internal balance between labile N and C in the shoot and root [46]. A higher investment of photosynthates to roots could maintain better supply of nutrients through a well-developed root system for eCO2 induced photosynthesis [47].”
- Line 326 Review citation mode in the entire text.
Response: Thanks, and the citation mode in the entire text were revised.
- Line 366-368 This section should be further elaborate.
Response: Thanks for your comment. We have added the elaborate “Nevertheless, nutrient use efficiency could be enhanced by each or all of the fol-lowing three pathways: (i) biomass increase and constant nutrient absorption, (ii) con-stant biomass and lower absorption of nutrients and (iii) increase in nutrient absorption rate is lower than that in dry mass increase rate [55]”.
- Line 424 Pink color does not always indicate N2 Please consider review this sentence and add further information.
Response: Thanks for your comment. The writings of “Although we did not measure how much N had been fixed under eCO2 and/or AMF, the observed pink color of fresh nodules, indicating the faba bean cultivated in this study had a certain degree of N2 fixation capacity.” has been changed to “However, it is still unknown that how much N could be fixed by N2-fixing bacteria under eCO2 and/or AMF. The determination of N2-fixing could thus provide the understanding of the dynamics behind the positive impacts of AMF and eCO2 on leguminous plants.”

Reviewer 3 Report
dear Authors, thanks for your manuscript. I have some suggestions:
title: I think that "facilitate" it's not clear.
I'm not a fan of abbreviation. I think that there are too much abbrevietions in your manuscript.
you can change daytime/nighttime into photoperiod of X h
concerning mycorrhization, I think that the appropiate method is trouvelot et al. 1986. it also provide degree of arbuscolarization that is an important parameter.
material and methods: change the design description. for example you wrote that the experiment involved two AMF, but there is only one AMF.
have test the presence of nodulation?
Author Response
- title: I think that "facilitate" it's not clear.
Response: Thanks for your comment. The title of “Arbuscular mycorrhizal colonization facilitated carbon, nitrogen, phosphorus and potassium accumulation in Vicia faba by modulating soil nutrient balance under elevated atmospheric CO2” has been change to “Arbuscular mycorrhization enhances nitrogen, phosphorus and potassium accumulation in Vicia faba by modulating soil nutrient balance under elevated CO2”
- I'm not a fan of abbreviation. I think that there are too much abbrevietions in your manuscript.
Response: Thanks for your concern. We would like to keep the common abbreviations of C, N, P, K for “carbon, nitrogen, phosphorous, potassium”, while have now used their full terms of “soil organic carbon, available nitrogen, available phosphorus and available potassium” for the previous short terms of “SOC, AN, AP and AK” in the revised version.
- you can change daytime/nighttime into photoperiod of X h
Response: Thanks for your comment. Daytime was from 07:00 a.m. to 19:00 p.m. and nighttime was from 19:00 p.m. to 07:00 a.m.
- concerning mycorrhization, I think that the appropiate method is trouvelot et al. 1986. it also provide degree of arbuscolarization that is an important parameter.
Response: Thanks for your suggestion. This study adopted the root AMF colonization determination method of Brundrett M, Bougher N, Dell B, Grove T, Malajczuk N. 1996. Working with Mycorrhizas in Forestry and Agriculture (Australian Centre for International Agricultural Research Monograph 32, 173-212), Canberra, Australia.
However, due to its availability and our Fresh language efficiency, we could not run the root AMF colonization determination method based on Trouvelot A, Kough JL, Gianinazzi-Pearson V, Gianinazi-Pearson V, Gianinazzi S. 1986. Mesure du taux de mycorhization VA d'un syst?me radiculaire. Recherche de methods d'estimation ayant une signification fonctionnelle. Physiology and Genetics Aspects of Mycorrhizae, 217-221, INRA, Paris.
- material and methods: change the design description. for example you wrote that the experiment involved two AMF, but there is only one AMF.
Response: Thanks for your concern, which is also commented by the reviewer 2. The previous sentence of “The experiment involved two AMF (with Funneliformis mosseae and without as control)” has been changed to “The experiment involved two AMF inoculations (Funneliformis mosseae, AMF and no F. mosseae as control, non-AMF)”.
- have test the presence of nodulation?
Response: Thanks for your comment and suggestion. We had just determined the nodule biomass production, and the nodulation and N2-fixing fixation would be considered in later experimentation.
